# Comparison of socio-economic determinants of COVID-19 testing and positivity in Canada: A multi-provincial analysis

Lilia Antonova[1,2], Chandy Somayaji[3], Jillian Cameron[3], Monica Sirski[4], Maria E. Sundaram[5,6], James Ted McDonald[7], Sharmistha Mishra[6,8,9], Jeffrey C. Kwong[6,10,11,12,13], Alan Katz[4,14], Stefan Baral[15], Lisa Caulley[1,2,16], Andrew Calzavara[6], Martin Corsten[17‡], Stephanie Johnson-Obaseki[1,2‡]*

1 Department of Otolaryngology–Head and Neck Surgery, University of Ottawa, Ottawa, ON, Canada, 2 Clinical Epidemiology Program, Ottawa Hospital Research Institute, Ottawa, ON, Canada, 3 New Brunswick Institute for Research, Data, and Training, University of New Brunswick, Fredericton, NB, Canada, 4 Manitoba Centre for Health Policy, University of Manitoba, Winnipeg, MB, Canada, 5 Center for Clinical Epidemiology and Population Health, Marshfield Clinic Research Institute, Marshfield, WI, United States of America, 6 ICES, Toronto, ON, Canada, 7 Department of Political Science, University of New Brunswick, Fredericton, NB, Canada, 8 Department of Medicine, University of Toronto, Toronto, ON, Canada, 9 MAP Center for Urban Health Solutions, St. Michael's Hospital, Unity Health Toronto, Toronto, ON, Canada, 10 Dalla Lana School of Public Health, University of Toronto, Toronto, ON, Canada, 11 Public Health Ontario, Toronto, Ontario, Canada, 12 Department of Family and Community Medicine, University of Toronto, Toronto, ON, Canada, 13 University Health Network, Toronto, ON, Canada, 14 Department of Community Health Sciences, University of Manitoba, Winnipeg, MB, Canada, 15 Department of Epidemiology, Johns Hopkins University School of Public Health, Baltimore, MD, United States of America, 16 Department of Epidemiology, Erasmus University Medical Center Rotterdam, Rotterdam, Netherlands, 17 Division of Otolaryngology–Head and Neck Surgery, Dalhousie University, Halifax, NS, Canada

‡ MC and SJO are co-senior authors on this work.
* stjohnson@toh.ca

## Abstract

### Background

The effects of the COVID-19 pandemic have been more pronounced for socially disadvantaged populations. We sought to determine how access to SARS-CoV-2 testing and the likelihood of testing positive for COVID-19 were associated with demographic factors, socioeconomic status (SES) and social determinants of health (SDH) in three Canadian provinces.

### Methods

An observational population-based cross-sectional study was conducted for the provinces of Ontario, Manitoba and New Brunswick between March 1, 2020 and April 27, 2021, using provincial health administrative data. After excluding residents of long-term care homes, those without current provincial health insurance and those who were tested for COVID-19 out of province, records from provincial healthcare administrative databases were reviewed for 16,900,661 healthcare users. Data was modelled separately for each province in accordance to a prespecified protocol and follow-up consultations among provincial statisticians

**Data Availability Statement:** The dataset from this study is held securely in coded form at provincial

HADs. While legal data sharing agreements between HADs and data providers (e.g., healthcare organizations and government) prohibit the respective HADs from making the dataset publicly available. Access may be granted to those who meet prespecified criteria for confidential access (for the province of Ontario available at www.ices. on.ca/DAS [email: das@ices.on.ca]). Instructions for submitting a data request can be found at the following link: https://www.ices.on.ca/DAS/ Submitting-your-request. Please note that the e-mail link found at the website under the data request link above, leads to the above-mentioned e-mail address: das@ices.on.ca.

**Funding:** This work was supported by Public Health Ontario. This study was also supported by ICES, which is funded by an annual grant from the Ontario Ministry of Health (MOH) and the Ministry of Long-Term Care (MLTC). This document used data adapted from the Statistics Canada Postal CodeOM Conversion File, which is based on data licensed from Canada Post Corporation, and/or data adapted from the Ontario Ministry of Health Postal Code Conversion File, which contains data copied under license from ©Canada Post Corporation and Statistics Canada. This study was supported by the Ontario Health Data Platform (OHDP), a Province of Ontario initiative to support Ontario's ongoing response to COVID-19 and its related impacts. The analyses for Ontario and Manitoba were supported by the Canadian Institutes of Health Research (VR5-172683). This work was supported by the Department of Health and Vitalité Health Network of the Province of New Brunswick under a contract with the New Brunswick Institute for Research, Data and Training at the University of New Brunswick. The results and conclusions are those of the authors and no official endorsement by the Government of New Brunswick or Vitalité Health Network was intended or should be inferred. The study sponsors did not participate in the design and conduct of the study; collection, management, analysis and interpretation of the data; preparation, review or approval of the manuscript; or the decision to submit the manuscript for publication. Parts of this material are based on data and/or information compiled and provided by the Canadian Institute for Health Information (CIHI) and by Ontario Health (OH). However, the analyses, conclusions, opinions and statements expressed herein are solely those of the authors, and do not reflect those of the funding or data sources; no endorsement by ICES, MOH, MLTC, OHDP, its partners, the Province of Ontario, CIHI or OH is intended or should be inferred. J.C.K. is supported by a Clinician-Scientist Award from the University of

and collaborators. We employed univariate and multivariate regression models to examine determinants of testing and test results.

## Results

After adjustment for other variables, female sex and urban residency were positively associated with testing, while female sex was negatively associated with test positivity. In New Brunswick and Ontario, individuals living in higher income areas were more likely to be tested, whereas in Manitoba higher income was negatively associated with both testing and positivity. High ethnocultural composition was associated with lower testing rates. Both high ethnocultural composition and high situational vulnerability increased the odds of testing positive for SARS-CoV-2.

## Discussion

We observed that multiple demographic, income and SDH factors were associated with SARS-CoV-2 testing and test positivity. Barriers to healthcare access identified in this study specifically relate to COVID-19 testing but may reflect broader inequities for certain at-risk groups.

## Introduction

The COVID-19 pandemic has highlighted the disproportionate vulnerability of certain populations to unanticipated health disasters. Even in high-income countries such as the United Kingdom and the United States increased positivity rates and poorer health outcomes, including death, from COVID-19 infection have been identified in populations with lower SES [1–4].

Previous work has identified some of the social determinants of health (SDH) contributing to socioeconomic differences in SARS-CoV-2 testing and infection rates in the United States, including household density, income, race, essential work, and health insurance status [5–8]. As health coverage was found to be an important factor associated with COVID-19 positivity in the United States, it is likely that the relative contribution of SDH and SES may be unique in a country such as Canada, where there is free access to universal healthcare coverage. Additionally, as public health measures and health care delivery are largely provincial responsibilities in Canada, the relative effects of SDH and SES may vary by province.

An important determinant of COVID-19 outcomes is access to SARS-CoV-2 testing [9, 10]. Equity in testing access contributes to timely disease management, helping to improve disease outcomes and to lighten the burden on the healthcare system [9, 11, 12]. An optimal testing strategy would make SARS-CoV-2 testing more available for socially disadvantaged groups to address the higher rates of infection in these populations. SES plays a role in SARS-CoV-2 testing disparities, with the most SARS-CoV-2-positive communities exhibiting some of the lowest testing rates in the United States [13, 14]. A similar relationship between high-risk communities and lower testing rates was demonstrated in Ontario, Canada [15]. The generalizability of this finding across various Canadian provinces remains a research question of interest.

In order to assess COVID-19 healthcare access and vulnerability across Canada, we sought to add to previous Canadian findings [15] by evaluating differences in SARS-CoV-2 testing rates and test results according to demographic, medical history, SES and SDH variables. Data

Toronto Department of Family and Community Medicine. SM is supported by a Tier 2 Canada Research Chair in Mathematical Modeling and Program Science. The funders had no role in study design, data collection and analysis, decision to publish, or preparation of the manuscript.

**Competing interests:** The authors have declared that no completing interests exist.

was obtained from census and administrative healthcare databases from the Canadian provinces of New Brunswick, Manitoba, and Ontario.

## Materials and methods

### Ethics approval

Ethics approval for this study was granted by the University of New Brunswick Research Ethics Board (New Brunswick), Ottawa Health Science Network Research Ethics Board (Ontario) and the University of Manitoba Health Research Ethics Board (Manitoba). The Manitoba ethics application was reviewed for data privacy and approved by the Government of Manitoba's Health Information and Privacy Committee (File No. 2020/2021-32).

### Study design and study population

We conducted a population-based, cross-sectional, observational study across three Canadian provinces (New Brunswick [NB], Manitoba [MB], and Ontario [ON]), which differ in their sizes and population distributions, as well as in the relative impact of COVID-19. In particular, MB has a large Northern community outside of Winnipeg. NB on the other hand is largely rural, without an obvious 'central hub' city. Additionally, the Eastern Provinces had a different approach to minimizing the impact of COVID-19 on the community by imposing a travel ban to and from all other provinces, which produce a delay in the waves of COVID-19 and the associated strains on testing capacity. In the early phases of the pandemic, access to testing was not widely available. Therefore, testing in all 3 provinces was limited largely to high-risk individuals, healthcare and personal care workers. For the majority of the time period of the study, all residents were eligible for SARS-CoV-2 testing in all provinces.

We used laboratory and health administrative databases (described under Data below and in S1 Table) in each province as well as area-level SES and SDH characteristics derived from census data. Data were de-identified prior to linkage.

The cohort in each province included all individuals registered with their respective provincial health system living as of March 1, 2020. The study period was from March 1, 2020 to April 27, 2021, which encompasses data from the start of the pandemic in Canada until the latest data available at the start of data collection. This period also precedes the shift towards antigen rapid-testing, the results of which are not collected in provincial healthcare databases. Long-term care residents, current residents without a valid healthcare card for the period of interest and out-of-province residents who accessed SARS-CoV-2 testing were excluded. Long-term care residents represent a population with a unique access to testing and vaccination in the course of the pandemic, which precludes their comparison to the general population.

### Data

Testing and test result information was identified from provincial databases (PCR test results) (S1 Table). Manitoba data were accessed from the Manitoba Population Research Repository housed at the Manitoba Centre for Health Policy, New Brunswick data were accessed from the New Brunswick Institute for Research, Data, and Training (NB-IRDT), and Ontario data were linked using unique encoded identifiers and analyzed at ICES (formerly the Institute for Clinical Evaluative Sciences). We evaluated test positivity as follows: first we considered whether an individual was tested; then, of the tested individuals, we considered if their test result was positive or negative. If an individual had more than one test in the time period of interest, we selected the first positive test or, if all tests were negative, the first negative test.

To explore factors contributing to testing and test positivity, we examined the role of relevant demographic, medical history, socioeconomic and SDH variables determined from provincial health administrative data (S1 Table). *Demographic variables* examined were age, sex and urban vs rural place of residence. Information about these variables was obtained from citizen databases containing healthcare information (S1 Table). Demographic data within these databases has been found to be 97% consistent with medical chart demographic information for Ontario [16] and more than 97% for Manitoba [17]. *Medical history variables* included the number of hospital admissions in the 3 years prior to the time period of interest (reflecting underlying health status) and comorbidities for which data was available for all provinces, including chronic obstructive pulmonary disease, hypertension, and diabetes. Comorbidities for which data was available only for some provinces (cancer [NB and ON], asthma [MB and ON], dementia and frailty [MB and ON], heart disease [MB and ON]), were only included in supplementary analyses (shown under Supplemental Data). Hospital admissions (0, 1, 2 or $\geq 3$) and comorbidities were defined using hospital discharge abstracts and physician billing claims. *Variables indicative of SES* included area-level quintiles for income (after tax income per person equivalent). *Variables indicative of SDH* included four measures from the Canadian Index of Multiple Deprivation (CIMD). Area-level variables were obtained by linking postal code to dissemination areas using the Postal Code Conversion File (version 7B for NB and ON, and 7D for MB), as described at https://www150.statcan.gc.ca/n1/pub/45-20-0001/ 452000012019002-eng.htm. The CIMD is a neighborhood-level index based on census data introduced by Statistics Canada in 2016 [18]. There are four SDH variables evaluated in the CIMD: residential instability (for example, the proportion of individuals who have moved in the past 5 years and the number of apartment buildings in the neighborhood); economic dependency (for example, the proportion of the population who are employed and the proportion receiving government assistance); ethnocultural composition (for example, the proportion of the population who identify as visible minorities or are recent immigrants); and situational vulnerability (for example, the proportion of population who are Aboriginal or who have not attained a high school diploma). All four components of the CIMD are ranked in levels from 1 to 5, with higher ranking indicating a greater presence of that component. We used the national level values, which encompass national level proportions for different indicators (as described at https://www150.statcan.gc.ca/n1/pub/45-20-0001/452000012019002-eng. htm) for all the indices to allow for comparison between provinces.

## Analysis

Each province undertook their own data analyses using standard variable definitions and protocol training, as well as regularly scheduled retraining sessions to ensure consistency. As data could only be analyzed intra-provincially due to health administrative database regulations and could not be combined between provinces for analysis, independent variables were selected *a priori* based on modelling results obtained in our previous study from Ontario [15]. We examined two separate outcomes: testing and test positivity and therefore we conducted parallel analyses for comparing a) tested versus untested individuals and b) individuals who tested positive versus those who tested negative. All analyses were conducted at the individual level, examining the effects of the demographic and neighborhood variables on each outcome variable for each province. A priori covariate selection was based on previous research conducted by our team [15]. Collinearity between the included variables was assessed in this previous investigation for data from Ontario and was not found for any of the variables. Collinearity was also assessed in the current study for data from Manitoba and was not observed for any variables. Descriptive statistics were calculated for all demographic variables,

SES and SDH. Statistical differences between demographic variables were evaluated using Welch's t-test for continuous variables and Fisher's chi-squared test for categorical variables. To examine unadjusted determinants of testing and test result, we conducted univariate logistic regression models on each variable. We also analyzed the variables in fully adjusted multivariate models with testing rates and test positivity as the dependent variables, according to the following logistic regression models:

Log odds of being tested OR positive for SARS-CoV-2 = $\beta_0$ + $\beta_1$(age group) + $\beta_2$(sex) + $\beta_3$(income quintile) + $\beta_4$(rurality) + $\beta_5$(number of hospital admissions in past year) + $\beta_6$(COPD) + $\beta_7$(hypertension) + $\beta_8$(diabetes) + $\beta_9$(residential instability quintile) + $\beta_{10}$(economic dependency quintile) + $\beta_{11}$(ethnocultural composition quintile) + β12(situational vulnerability quintile).

Variables for which data were not available in all provinces were not included in the main models in order to allow for side-to-side comparisons between provinces. However, they can be found in additional multiple regression analyses presented under Supplemental Data for reference. Statistical analyses were conducted using SAS v 9.4 (SAS 2013 Statistical Analysis Software, SAS Institute Inc, Cary, NC).

## Results

Data from provincial surveillance databases indicated that many individuals enrolled in the provincial health insurance systems received testing over the study period: 25% for NB (N = 159,150), 40% for MB (N = 401,251) and 49% for ON (N = 4,792,449) (Fig 1). Of those tested, 1% (N = 1645), 9% (N = 35,458) and 5% (N = 258,769), tested positive for SARS-CoV-2 (NB, MB and ON, respectively) (Fig 1). Comparison of the tested and untested populations demonstrated that they differed in the distribution of age, sex and place of residence (Table 1) (p<0.001 for all variables for all provinces). The same was found for those who tested positive, as compared to those who tested negative (Table 1) (p<0.0001 for all variables for all provinces).

In Manitoba and Ontario, the 20–34 and 35–**49 age** groups were most likely to be tested (Table 2, Fig 2), while children aged 0–4 showed the highest **odds** of **being** tested in New Brunswick. Children aged 5–19 were most likely to test positive in Manitoba and Ontario, whereas the 85+ age group was associated with the highest test positivity in New Brunswick (Table 3, Fig 3). In all provinces, females were significantly more likely to be tested than males and less likely to test positive (Tables 2 and 3, Figs 2 and 3). Finally, urban residents were significantly more likely to be tested in all provinces, and more likely to test positive in New Brunswick and Ontario (Tables 2 and 3, Figs 2 and 3).

Having more than one hospitalization in the 3 years prior to the start of the study increased the odds of being tested (Table 2, Fig 2). Conversely, having at least one hospitalization corresponded to a decreased likelihood of testing positive (Table 3, Fig 3). In all provinces, COPD, hypertension and diabetes were associated with a higher rate of testing (Table 2, Fig 2). Hypertension and diabetes also increased the odds of testing positive for SARS-CoV-2 (Table 3, Fig 3). Interestingly, having COPD conferred a lower likelihood of testing positive.

Residing in higher income areas was positively associated with testing in New Brunswick and Ontario, but negatively associated with testing in Manitoba (Table 2, Fig 2). Higher neighbourhood income was associated with significantly lower test positivity in Manitoba and Ontario (Table 3, Fig 3).

Areas with highest ethnocultural composition exhibited lower testing rates (Table 2, Fig 2). Both high ethnocultural composition and high situational vulnerability were associated with

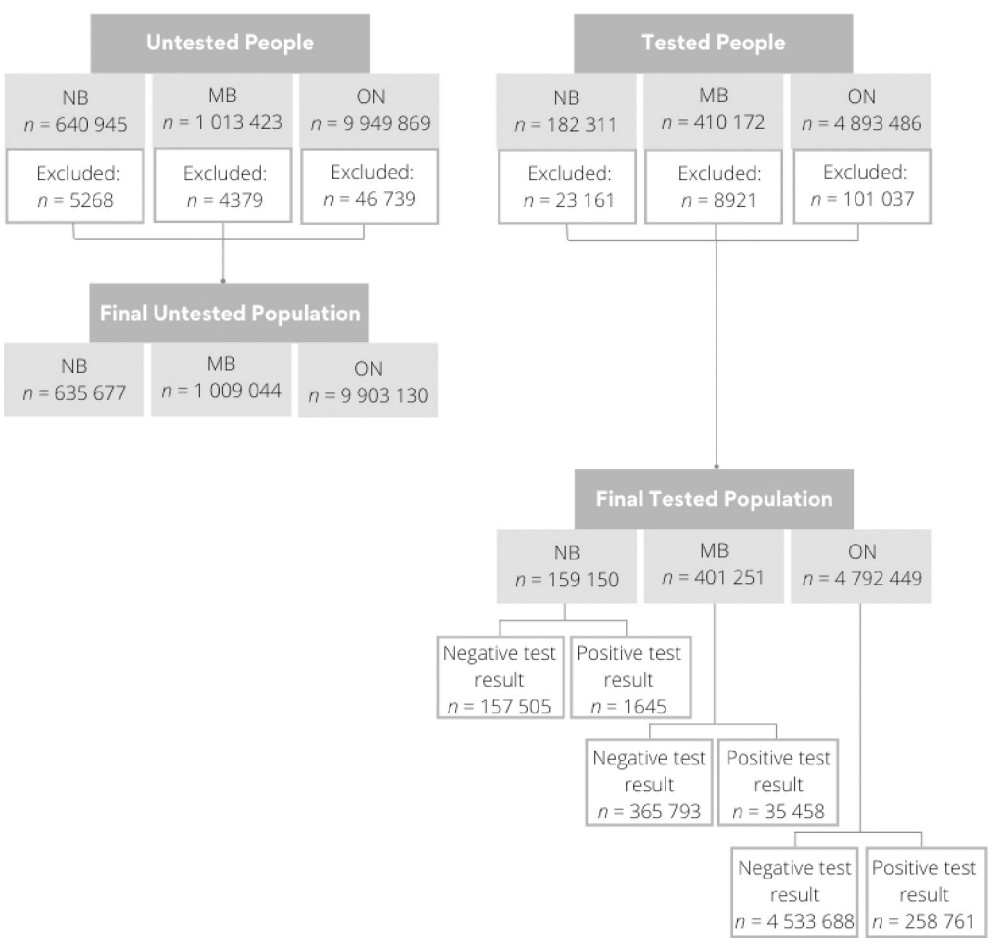

**Fig 1. Flow diagram for untested and COVID-19 tested groups, and negative or positive test results for the provinces of New Brunswick (NB), Manitoba (MB), and Ontario (ON).**

increased odds of testing positive, whereas residential instability was associated with lower test positivity (Table 3, Fig 3).

## Discussion

In this work we accessed the unique opportunity of having systematically recorded testing and test result data for a pandemic in progress in order to identify disparities in socioeconomic vulnerability to health emergencies and inequities in access to healthcare. Specifically, we sought to examine differences between SARS-CoV-2 testing and test positivity rates between three Canadian provinces, as well as the demographic, medical history, SES and SDH variables that may contribute to such differences. We observed that both testing rates and test positivity rates varied significantly among provinces.

The COVID-19 pandemic affected Canadian provinces to a varying extent and saw contrasting governmental strategies to contain it. New Brunswick was part of the "Atlantic Bubble", a group of 4 provinces that severely restricted travel external to the bubble during the pandemic, while Manitoba and Ontario had less severe restrictions. This likely decreased the burden of COVID-19 in New Brunswick during the first three waves of the pandemic. Data released by Oxford's Blavatnik School of Government regarding provincial Covid-19

**Table 1. Population distribution according to demographic characteristics and social determinants of health.**

| | New Brunswick | | | | Manitoba | | | | Ontario | | | |
|---|---|---|---|---|---|---|---|---|---|---|---|---|
| | All individuals | | Tested individuals | | All individuals | | Tested individuals | | All individuals | | Tested individuals | |
| | Untested control | All tested | Tested negative | Tested positive | Untested control | All tested | Tested negative | Tested positive | Untested control | All tested | Tested negative | Tested positive |
| N | 635 677 | 159 150 | 157 505 | 1645 | 1 009 004 | 401 251 | 365 793 | 35 458 | 9 903 130 | 4 792 449 | 4 533 688 | 258 761 |
| Age: mean (SD) | 44.81 (23.61) | 41.11 (22.85) | 41.05 (22.84) | 46.00 (22.84) | 39.27 (23.87) | 38.89 (22.47) | 38.90 (22.44) | 38.54 (22.81) | 41.09 (23.25) | 40.16 (22.13) | 40.21 (22.24) | 39.19 (20.22) |
| Ages: 0–4 (%) | 4.02 | 5.21 | 5.24 | 2.81 | 6.56 | 4.94 | 5.00 | 4.35 | 5.27 | 4.53 | 4.62 | 3.00 |
| Ages: 5–19 (%) | 14.74 | 15.22 | 15.26 | 11.53 | 19.33 | 16.65 | 16.49 | 18.29 | 17.25 | 15.06 | 15.02 | 15.86 |
| Ages: 20–34 (%) | 16.49 | 20.11 | 20.12 | 18.66 | 19.12 | 24.41 | 24.31 | 25.51 | 18.56 | 23.69 | 23.62 | 24.92 |
| Ages: 35–49 (%) | 18.02 | 21.32 | 21.33 | 20.55 | 18.48 | 21.84 | 21.98 | 20.35 | 19.41 | 21.21 | 21.10 | 23.22 |
| Ages: 50–64 (%) | 22.92 | 20.84 | 20.79 | 25.81 | 18.92 | 17.59 | 17.61 | 17.36 | 21.06 | 20.43 | 20.34 | 22.08 |
| Ages: 65–74 (%) | 13.98 | 9.97 | 9.97 | 9.64 | 10.27 | 7.59 | 7.72 | 6.24 | 10.91 | 8.34 | 8.44 | 6.63 |
| Ages: 75–84 (%) | 6.99 | 5.00 | 5.00 | 5.78 | 5.08 | 4.21 | 4.23 | 3.93 | 5.63 | 4.39 | 4.47 | 2.99 |
| Ages: 85+ (%) | 2.84 | 2.32 | 2.29 | 5.37 | 2.22 | 2.78 | 2.66 | 3.97 | 1.91 | 2.35 | 2.41 | 1.31 |
| Males (%) | 50.69 | 44.17 | 44.14 | 46.67 | 51.20 | 46.08 | 45.79 | 49.05 | 50.63 | 46.28 | 45.96 | 51.91 |
| Females (%) | 49.31 | 55.83 | 55.86 | 53.33 | 48.80 | 53.92 | 54.21 | 50.95 | 49.37 | 53.72 | 54.05 | 48.09 |
| Rural residence (%) | 38.15 | 31.11 | 31.14 | 28.21 | 28.76 | 28.15 | 27.66 | 33.13 | 10.27 | 9.52 | 9.86 | 3.64 |
| Urban residence (%) | 61.85 | 68.89 | 68.86 | 71.79 | 71.24 | 71.85 | 72.34 | 66.87 | 89.73 | 90.48 | 90.14 | 96.36 |
| Household income quintile (%) | | | | | | | | | | | | |
| 1 (lowest) | 20.30 | 19.25 | 19.22 | 21.70 | 20.40 | 23.17 | 21.72 | 38.10 | 19.54 | 18.95 | 18.60 | 25.11 |
| 2 | 19.83 | 18.80 | 18.79 | 20.30 | 19.00 | 18.55 | 18.56 | 18.39 | 19.59 | 19.05 | 18.90 | 21.57 |
| 3 | 20.09 | 19.36 | 19.35 | 20.36 | 19.76 | 18.65 | 18.95 | 15.54 | 20.06 | 19.93 | 19.84 | 21.48 |
| 4 | 19.60 | 20.03 | 20.06 | 17.63 | 21.41 | 19.98 | 20.42 | 15.47 | 20.02 | 20.27 | 20.45 | 17.24 |
| 5 (highest) | 19.81 | 22.15 | 22.18 | 19.57 | 19.46 | 19.66 | 20.35 | 12.50 | 19.73 | 20.93 | 21.36 | 13.45 |
| Underlying chronic health conditions (%) | | | | | | | | | | | | |
| COPD | 7.14 | 7.69 | 7.68 | 8.09 | 0.36 | 0.81 | 0.79 | 0.95 | 1.74 | 2.35 | 2.42 | 1.18 |
| Hypertension | 25.69 | 22.67 | 22.61 | 28.09 | 15.94 | 16.42 | 16.29 | 17.68 | 21.06 | 20.55 | 20.58 | 20.08 |
| Diabetes | 10.51 | 10.04 | 10.01 | 12.77 | 6.88 | 8.73 | 8.44 | 11.74 | 9.99 | 10.17 | 10.04 | 12.52 |
| Hospital admissions, past 3 years: Mean (SD) | 1.86 (1.45) | 2.03 (1.83) | 2.03 (1.83) | 2.17 (1.85) | 0.19 (0.61) | 0.34 (0.98) | 0.33 (0.97) | 0.39 (1.13) | 0.17 (0.54) | 0.26 (0.85) | 0.27 (0.87) | 0.17 (0.60) |
| CIMD–Residential instability quintiles (%) | N = 631 440 | N = 158 015 | N = 156 380 | N = 1635 | N = 995 798 | N = 395 783 | N = 360 782 | N = 35 001 | N = 9 791 430 | N = 4 747 784 | N = 4 483 422 | N = 255 665 |
| 1 (least) | 21.40 | 21.82 | 21.84 | 20.06 | 21.73 | 23.02 | 22.02 | 33.36 | 20.89 | 20.56 | 20.71 | 17.86 |
| 2 | 25.64 | 24.29 | 24.31 | 22.32 | 17.78 | 17.30 | 17.62 | 13.99 | 23.12 | 22.61 | 22.62 | 22.45 |
| 3 | 22.68 | 22.71 | 22.70 | 23.30 | 15.70 | 15.89 | 16.16 | 13.17 | 19.25 | 19.46 | 19.44 | 19.80 |
| 4 | 16.60 | 17.63 | 17.61 | 19.33 | 20.26 | 18.67 | 19.03 | 14.95 | 16.71 | 17.39 | 17.46 | 16.11 |
| 5 (most) | 13.68 | 13.55 | 13.54 | 14.98 | 24.53 | 25.12 | 25.17 | 24.55 | 20.03 | 19.99 | 19.77 | 23.78 |
| CIMD–Economic dependency quintiles (%) | | | | | | | | | | | | |
| 1 (least) | 11.96 | 13.42 | 13.49 | 6.61 | 18.51 | 18.92 | 18.73 | 20.88 | 22.46 | 23.11 | 23.16 | 22.15 |
| 2 | 14.15 | 14.84 | 14.87 | 11.74 | 20.24 | 20.67 | 20.69 | 20.44 | 22.73 | 22.97 | 22.81 | 25.86 |
| 3 | 19.25 | 19.22 | 19.22 | 19.39 | 19.02 | 19.82 | 19.67 | 21.29 | 21.21 | 21.13 | 21.09 | 21.80 |
| 4 | 23.28 | 23.08 | 23.00 | 30.70 | 19.29 | 20.12 | 19.96 | 21.83 | 18.66 | 18.28 | 18.24 | 19.02 |

*(Continued)*

**Table 1.** (Continued)

| | New Brunswick | | | | Manitoba | | | | Ontario | | | |
|---|---|---|---|---|---|---|---|---|---|---|---|---|
| | All individuals | | Tested individuals | | All individuals | | Tested individuals | | All individuals | | Tested individuals | |
| | Untested control | All tested | Tested negative | Tested positive | Untested control | All tested | Tested negative | Tested positive | Untested control | All tested | Tested negative | Tested positive |
| 5 (most) | 31.35 | 29.44 | 29.42 | 31.56 | 22.93 | 20.47 | 20.95 | 15.55 | 14.94 | 14.51 | 14.70 | 11.17 |
| CIMD–Ethnocultural composition (%) | | | | | | | | | | | | |
| 1 (least) | 44.07 | 44.33 | 44.28 | 48.69 | 13.61 | 13.02 | 13.26 | 10.53 | 7.34 | 7.28 | 7.55 | 2.54 |
| 2 | 36.10 | 35.22 | 35.27 | 30.76 | 19.62 | 20.57 | 20.74 | 18.91 | 15.41 | 15.48 | 16.00 | 6.36 |
| 3 | 12.72 | 13.33 | 13.33 | 13.27 | 18.97 | 20.93 | 20.97 | 20.49 | 18.80 | 19.81 | 20.30 | 11.07 |
| 4 | 4.94 | 5.21 | 5.21 | 5.14 | 20.86 | 20.83 | 20.93 | 19.88 | 22.22 | 23.32 | 23.54 | 19.43 |
| 5 (most) | 2.16 | 1.91 | 1.91 | 2.14 | 26.93 | 24.65 | 24.11 | 30.20 | 36.23 | 34.11 | 32.61 | 60.60 |
| CIMD–Situational vulnerability (%) | | | | | | | | | | | | |
| 1 (least) | 16.32 | 18.97 | 19.02 | 14.25 | 13.93 | 14.00 | 14.47 | 9.15 | 30.27 | 30.74 | 31.04 | 25.38 |
| 2 | 16.76 | 16.96 | 17.02 | 11.68 | 20.02 | 19.14 | 19.67 | 13.59 | 22.3 | 22.49 | 22.49 | 22.51 |
| 3 | 14.97 | 15.68 | 15.67 | 17.00 | 16.68 | 15.23 | 15.44 | 13.10 | 18.03 | 17.98 | 17.93 | 18.74 |
| 4 | 24.10 | 23.55 | 23.46 | 32.11 | 20.32 | 19.05 | 19.24 | 17.11 | 15.39 | 15.15 | 15.08 | 16.39 |
| 5 (most) | 27.84 | 24.84 | 24.84 | 24.95 | 29.06 | 32.57 | 31.17 | 47.05 | 14.01 | 13.64 | 13.45 | 16.97 |

responses across Canada [19] for the year 2020 indicates that stringency measures across all provinces strongly correlated with SARS-CoV-2 positivity rates and that among examined provinces in this study, Covid-19 measures can be ranked as most stringent in New Brunswick (these were the most stringent measures across all Canadian provinces) and least stringent in Manitoba. Our results indicate that the level of stringency was indeed inversely correlated with test positivity for the examined time period. In addition, broader SARS-CoV-2 testing criteria for children than those in other provinces were introduced in New Brunswick in April, 2020, which we observed reflected in the high testing rate for the youngest demographic in this province. There also exist significant differences between the provinces' distributions of population place of residence and socio-ethnocultural composition, factors which we found to be important for determining SARS-CoV-2 testing and test positivity.

We examined the possible confounding effects of pre-existing health status and comorbid conditions on SARS-CoV-2 testing and positivity. COPD was associated with higher testing rates and lower test positivity rates. One possible explanation for this finding is that COPD has symptoms that mimic SARS-CoV-2 infection and may prompt SARS-CoV-2 testing more frequently. Patients with COPD are also known to be at a higher risk of severe disease and associated mortality [20], which may result in closer monitoring for COVID-19 infection. Diabetes and hypertension have both been linked to developing serious illness due to SARS-CoV-2 infection [21, 22]. Here we observed that the association between these comorbidities and COVID-19 is reflected in higher test positivity rates. This may be attributable to an interplay of factors, including immune dysfunction due to hyperglycemia in diabetic patients [23] and increase of the SARS-CoV-2 binding protein ACE2 in patients with hypertension or diabetes that are treated with ACE inhibitors and/or angiotensin II type-1 receptor blockers [24].

Place of residence was found to play an important role in SARS-CoV-2 testing, with urban residents being more likely to be tested in all three provinces. This parallels what is previously known about healthcare testing and capacity for other health conditions and has been demonstrated for SARS-CoV-2 testing by Huang and colleagues [25]. Reasons likely include resource

**Table 2. Odds ratios and 95% confidence intervals for multivariable regression models of the odds of SARS-CoV-2 testing in New Brunswick (NB), Manitoba (MB), and Ontario (ON).**

| Variable | NB | MB | ON |
|---|---|---|---|
| Age group: 5–19 vs 0–4 | **0.78 (0.75, 0.80)** | **1.40 (1.37, 1.43)** | **1.03 (1.03, 1.04)** |
| Age group: 20–34 vs 0–4 | **0.89 (0.86, 0.91)** | **1.98 (1.95, 2.02)** | **1.47 (1.46, 1.47)** |
| Age group: 35–49 vs 0–4 | **0.83 (0.81, 0.86)** | **1.83 (1.79, 1.86)** | **1.24 (1.23, 1.24)** |
| Age group: 50–64 vs 0–4 | **0.61 (0.59, 0.62)** | **1.36 (1.33, 1.38)** | **1.05 (1.05, 1.05)** |
| Age group: 65–74 vs 0–4 | **0.42 (0.41, 0.44)** | **0.97 (0.95, 0.99)** | **0.76 (0.75, 0.76)** |
| Age group: 75–84 vs 0–4 | **0.38 (0.36, 0.39)** | **0.92 (0.90, 0.94)** | **0.71 (0.71, 0.72)** |
| Age group: 85+ vs 0–4 | **0.40 (0.39, 0.42)** | 0.99 (0.95, 1.02) | **1.04 (1.04, 1.05)** |
| Sex: Female vs Male | **1.28 (1.27, 1.30)** | **1.20 (1.20, 1.21)** | **1.09 (1.09, 1.09)** |
| Income quintile: 2 vs 1 | 1.02 (1.00, 1.04) | **0.90 (0.89, 0.91)** | **0.94 (0.94, 0.94)** |
| Income quintile: 3 vs 1 | **1.08 (1.06, 1.11)** | **0.87 (0.86, 0.89)** | **0.997 (0.995, 0.999)** |
| Income quintile: 4 vs 1 | **1.20 (1.17, 1.23)** | **0.87 (0.86, 0.89)** | **1.05 (1.04, 1.05)** |
| Income quintile: 5 vs 1 | **1.38 (1.34, 1.42)** | **0.87 (0.85, 0.88)** | **1.12 (1.11, 1.12)** |
| Rurality: Urban vs Rural | **1.45 (1.43, 1.47)** | **1.12 (1.11, 1.14)** | **1.05 (1.05, 1.05)** |
| Hospital admissions: 1 vs 0 | **1.37 (1.35, 1.39)** | **1.41 (1.39, 1.42)** | **0.84 (0.83, 0.84)** |
| Hospital admissions: 2 vs 0 | **1.52 (1.49, 1.56)** | **1.67 (1.63, 1.70)** | **1.06 (1.05, 1.07)** |
| Hospital admissions: ≥3 vs 0 | **1.87 (1.82, 1.92)** | **2.44 (2.37, 2.51)** | **1.74 (1.72, 1.75)** |
| Comorbidities: COPD | **1.34 (1.31, 1.37)** | **1.60 (1.51, 1.69)** | **1.38 (1.37, 1.39)** |
| Comorbidities: hypertension | **1.08 (1.06, 1.09)** | **1.12 (1.11, 1.14)** | **1.10 (1.10, 1.11)** |
| Comorbidities: diabetes | **1.11 (1.09, 1.13)** | **1.30 (1.28, 1.32)** | **1.10 (1.09, 1.10)** |
| CIMD Residential instability: 2 vs 1 | **1.04 (1.03, 1.06)** | **0.93 (0.92, 0.94)** | **0.95 (0.95, 0.96)** |
| CIMD Residential instability: 3 vs 1 | **1.15 (1.13, 1.17)** | **0.97 (0.96, 0.99)** | **1.01 (1.01, 1.01)** |
| CIMD Residential instability: 4 vs 1 | **1.23 (1.20, 1.25)** | **0.91 (0.89, 0.92)** | **1.07 (1.06, 1.07)** |
| CIMD Residential instability: 5 vs 1 | **1.11 (1.08, 1.14)** | **0.94 (0.93, 0.95)** | **1.04 (1.04, 1.04)** |
| CIMD Economic dependency: 2 vs 1 | 1.01 (0.99, 1.03) | **0.97 (0.95, 0.98)** | **1.01 (1.01, 1.01)** |
| CIMD Economic dependency: 3 vs 1 | 1.02 (0.99, 1.04) | **1.01 (1.00, 1.03)** | 1.00 (1.00, 1.00) |
| CIMD Economic dependency: 4 vs 1 | **1.08 (1.05, 1.10)** | 0.99 (0.98, 1.01) | **0.994 (0.992, 0.997)** |
| CIMD Economic dependency: 5 vs 1 | **1.11 (1.09, 1.14)** | **0.87 (0.86, 0.88)** | 0.999 (0.996, 1.001) |
| CIMD Ethnocultural composition: 2 vs 1 | **0.95 (0.94, 0.97)** | 1.01 (0.99, 1.02) | **1.019 (1.017, 1.022)** |
| CIMD Ethnocultural composition: 3 vs 1 | **0.93 (0.92, 0.96)** | **1.03 (1.02, 1.05)** | **1.028 (1.026, 1.031)** |
| CIMD Ethnocultural composition: 4 vs 1 | **0.90 (0.88, 0.93)** | **0.90 (0.88, 0.91)** | **1.006 (1.004, 1.009)** |
| CIMD Ethnocultural composition: 5 vs 1 | **0.77 (0.74, 0.81)** | **0.76 (0.75, 0.77)** | **0.92 (0.92, 0.92)** |
| CIMD Situational vulnerability: 2 vs 1 | **0.94 (0.93, 0.96)** | **0.97 (0.96, 0.99)** | **1.005 (1.002, 1.007)** |
| CIMD Situational vulnerability: 3 vs 1 | **1.05 (1.03, 1.08)** | **0.92 (0.91, 0.94)** | **1.01 (1.01, 1.01)** |
| CIMD Situational vulnerability: 4 vs 1 | **1.03 (1.01, 1.05)** | **0.92 (0.90, 0.93)** | **1.007 (1.004, 1.009)** |
| CIMD Situational vulnerability: 5 vs 1 | 1.02 (1.00, 1.05) | **1.03 (1.01, 1.05)** | **0.997 (0.994, 0.999)** |

Adjusted odds ratios and 95% CIs are presented. The adjusted model contains all variables listed in the table and does not contain additional covariates that are not listed in the table.

constraints that are magnified in rural areas, difficulty with transportation and cultural perceptions of health and healthcare [26–28]. As a significantly higher proportion of Ontario residents are urban-dwellers (85%) compared to the other two provinces, place of residence is an important contributor to the observed inter-province differences in testing. Interestingly, test positivity effects were less consistent among provinces–urban residents tested positive more frequently in Ontario and New Brunswick, but less frequently in Manitoba. This discrepancy may reflect a testing drive that was done after an early outbreak in a rural northern area of

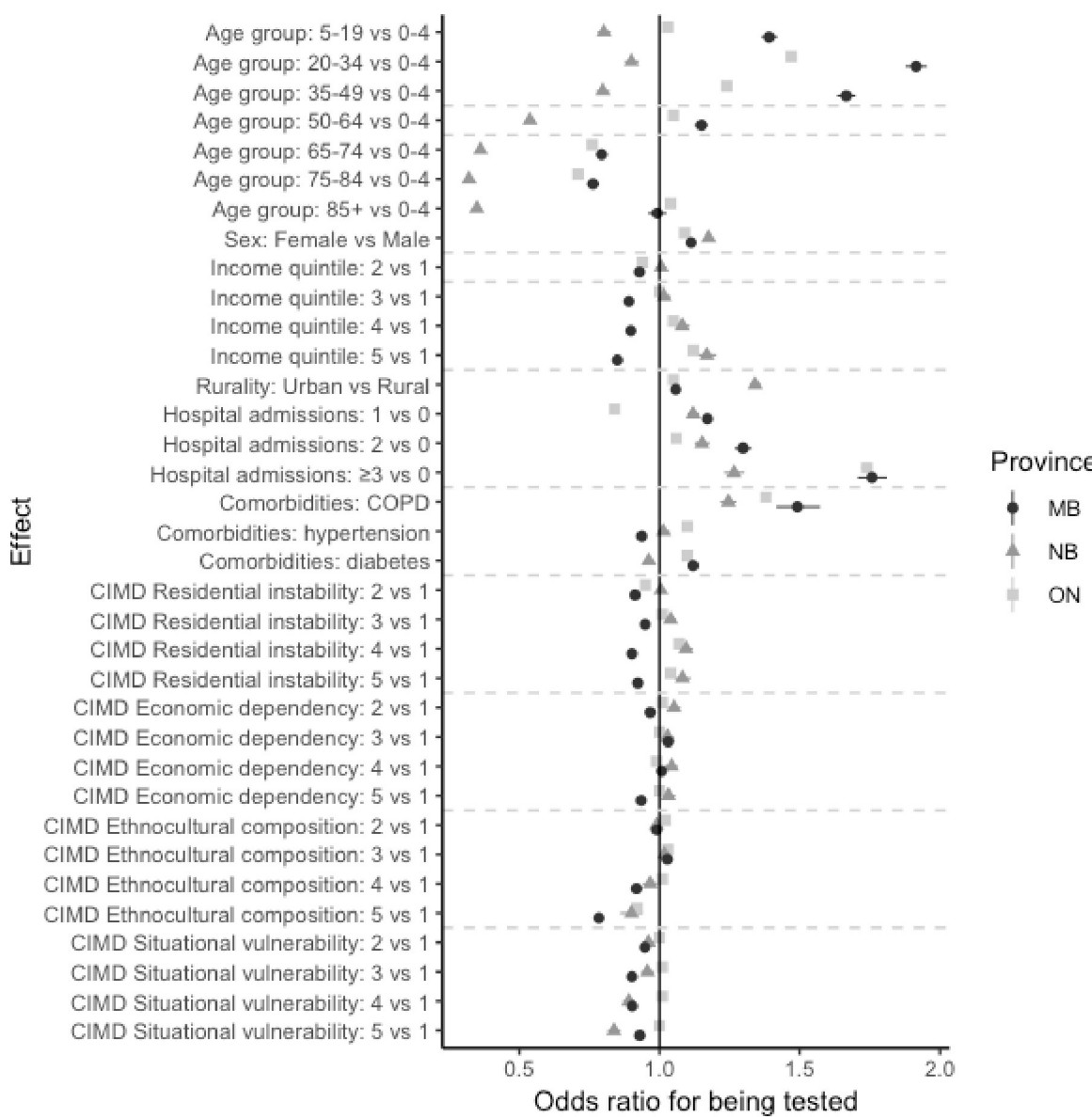

**Fig 2. Distribution of independent variables associations with being tested for SARS-CoV-2, multivariate models.**

Manitoba [29]. Differences in SARS-CoV-2 positivity between urban and rural areas could be explained by factors associated with population density [15, 30], such as differing levels of congestion in living conditions, occupational exposures, use of public transportation, and opportunity for social distancing.

Socioeconomic factors and SDH were found to play a significant role in SARS-CoV-2 testing and positivity. Higher income was correlated with higher rates of testing in Ontario and New Brunswick, while Manitoba was an outlier with highest rates of testing seen in the groups with the lowest SES. This may relate to the testing drive in Northern Manitoba referenced earlier, thus demonstrating the effectiveness of an equity-driven approach to testing in marginalized communities. The observed dependence of SARS-CoV-2 testing on income is an area of concern, as it may be a product of inequity in resource distribution and testing accessibility.

**Table 3. Odds ratios and confidence intervals for multivariable regression models of the odds of testing positive for SARS-CoV-2 (among tested individuals), in New Brunswick (NB), Manitoba (MB), and Ontario (ON).**

| Variable | NB | MB | ON |
|---|---|---|---|
| Age group: 5–19 vs 0–4 | **1.42 (1.02, 1.96)** | **1.34 (1.26, 1.42)** | **1.53 (1.49, 1.57)** |
| Age group: 20–34 vs 0–4 | **1.71 (1.25, 2.34)** | **1.29 (1.22, 1.37)** | **1.44 (1.40, 1.48)** |
| Age group: 35–49 vs 0–4 | **1.81 (1.33, 2.47)** | **1.17 (1.10, 1.25)** | **1.52 (1.48, 1.56)** |
| Age group: 50–64 vs 0–4 | **2.22 (1.63, 3.03)** | **1.21 (1.13, 1.29)** | **1.52 (1.48, 1.56)** |
| Age group: 65–74 vs 0–4 | **1.75 (1.24, 2.47)** | 0.95 (0.88, 1.02) | **1.19 (1.15, 1.22)** |
| Age group: 75–84 vs 0–4 | **2.16 (1.48, 3.14)** | 0.93 (0.85, 1.02) | **1.05 (1.01, 1.09)** |
| Age group: 85+ vs 0–4 | **4.42 (3.00, 6.50)** | **1.15 (1.03, 1.29)** | **0.90 (0.86, 0.94)** |
| Sex: Female vs Male | **0.90 (0.81, 0.99)** | **0.87 (0.85, 0.89)** | **0.81 (0.80, 0.81)** |
| Income quintile: 2 vs 1 | 0.96 (0.81, 1.14) | **0.68 (0.65, 0.70)** | **0.84 (0.83, 0.86)** |
| Income quintile: 3 vs 1 | 1.10 (0.91, 1.33) | **0.59 (0.56, 0.62)** | **0.77 (0.75, 0.78)** |
| Income quintile: 4 vs 1 | 1.04 (0.84, 1.28) | **0.55 (0.52, 0.57)** | **0.63 (0.62, 0.65)** |
| Income quintile: 5 vs 1 | 1.24 (0.99, 1.65) | **0.45 (0.43, 0.48)** | **0.55 (0.54, 0.57)** |
| Rurality: Urban vs Rural | **1.46 (1.29, 1.65)** | **0.76 (0.73, 0.79)** | **1.31 (1.28, 1.35)** |
| Hospital admissions: 1 vs 0 | **0.77 (0.66, 0.90)** | **0.96 (0.92, 0.99)** | **0.89 (0.88, 0.90)** |
| Hospital admissions: 2 vs 0 | **0.73 (0.59, 0.89)** | 0.94 (0.88, 1.00) | **0.75 (0.72, 0.77)** |
| Hospital admissions: ≥3 vs 0 | 0.83 (0.67, 1.03) | 1.04 (0.97, 1.12) | **0.50 (0.48, 0.53)** |
| Comorbidities: COPD | **0.82 (0.68, 1.00)** | 0.96 (0.83, 1.12) | **0.67 (0.64, 0.69)** |
| Comorbidities: hypertension | 1.06 (0.92, 1.22) | **1.09 (1.04, 1.13)** | **1.08 (1.06, 1.09)** |
| Comorbidities: diabetes | 1.12 (0.95, 1.31) | **1.27 (1.22, 1.33)** | **1.31 (1.29, 1.33)** |
| CIMD Residential instability: 2 vs 1 | 0.91 (0.78, 1.07) | **0.73 (0.70, 0.76)** | **0.94 (0.93, 0.95)** |
| CIMD Residential instability: 3 vs 1 | 1.09 (0.93, 1.28) | **0.67 (0.65, 0.70)** | **0.84 (0.82, 0.85)** |
| CIMD Residential instability: 4 vs 1 | 1.06 (0.89, 1.28) | **0.59 (0.56, 0.61)** | **0.67 (0.66, 0.69)** |
| CIMD Residential instability: 5 vs 1 | 0.98 (0.78, 1.24) | **0.53 (0.51, 0.55)** | **0.63 (0.62, 0.64)** |
| CIMD Economic dependency: 2 vs 1 | **1.60 (1.25, 2.05)** | **0.92 (0.89, 0.96)** | **1.06 (1.05, 1.07)** |
| CIMD Economic dependency: 3 vs 1 | **1.91 (1.51, 2.42)** | 1.02 (0.98, 1.06) | 0.99 (0.98, 1.01) |
| CIMD Economic dependency: 4 vs 1 | **2.55 (2.02, 3.23)** | **1.06 (1.02, 1.10)** | 1.00 (0.99, 1.01) |
| CIMD Economic dependency: 5 vs 1 | **2.04 (1.60, 2.59)** | **0.88 (0.84, 0.92)** | **0.91 (0.89, 0.92)** |
| CIMD Ethnocultural composition: 2 vs 1 | **0.85 (0.76, 0.95)** | **1.16 (1.11, 1.22)** | **1.18 (1.14, 1.21)** |
| CIMD Ethnocultural composition: 3 vs 1 | 1.11 (0.94, 1.30) | **1.48 (1.41, 1.56)** | **1.64 (1.59, 1.68)** |
| CIMD Ethnocultural composition: 4 vs 1 | 1.01 (0.80, 1.28) | **1.54 (1.46, 1.62)** | **2.50 (2.43, 2.57)** |
| CIMD Ethnocultural composition: 5 vs 1 | 1.15 (0.80, 1.65) | **1.73 (1.64, 1.82)** | **4.85 (4.72, 4.98)** |
| CIMD Situational vulnerability: 2 vs 1 | 0.85 (0.69, 1.04) | 1.04 (0.99, 1.09) | **1.16 (1.15, 1.18)** |
| CIMD Situational vulnerability: 3 vs 1 | **1.29 (1.05, 1.58)** | **1.21 (1.14, 1.28)** | **1.20 (1.18, 1.21)** |
| CIMD Situational vulnerability: 4 vs 1 | **1.60 (1.32, 1.95)** | **1.23 (1.16, 1.30)** | **1.20 (1.18, 1.22)** |
| CIMD Situational vulnerability: 5 vs 1 | **1.29 (1.04, 1.61)** | **1.38 (1.30, 1.47)** | **1.35 (1.33, 1.38)** |

Adjusted odds ratios and 95% CIs are presented. The adjusted model contains all variables listed in the table and does not contain additional covariates that are not listed in the table.

Lack of access to SARS-CoV-2 testing early in the disease course in highly impacted populations is known to contribute to poorer health outcomes [31].

Interestingly, higher residential instability was found to be associated with lower odds of testing positive for COVID-19. The CIMD defines the index of residential instability at the national level as accounting for measures such as the proportion of the population who have moved in the past five years, the proportion of persons living alone, and the proportion of occupied units that are rented rather than owned. Therefore, it is a proxy for neighborhood

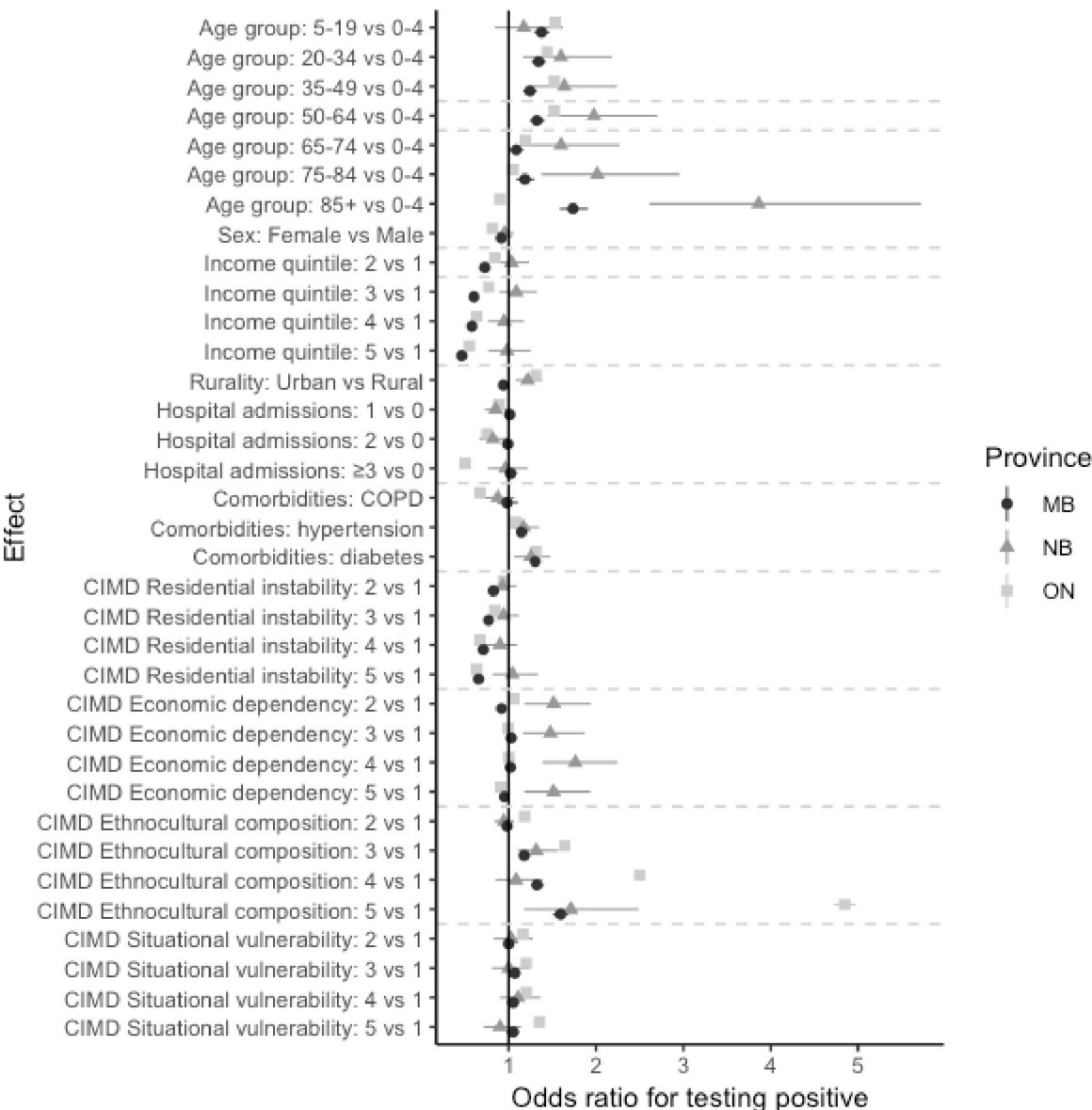

**Fig 3. Distribution of independent variables associations with testing positive for SARS-CoV-2, multivariate models.**

support and cohesiveness and household size. High levels of residential instability are likely to indicate a lower local community connection and a higher likelihood of living alone. Increased household size is, in turn, a well-established risk factor for COVID-19 infection [32]. Therefore, our findings confirm that lower interpersonal opportunities of exposure, at the home or in the community, reduce the changes of COVID-19 positivity.

Ethnocultural composition showed the strongest association to SARS-CoV-2 testing and testing outcomes out of the five CIMD index components, with more racialized or higher immigrant rate communities showing significantly lower testing rates and higher positivity rates. These disparities are consistent with previous observations from Canada [15, 33] and the United States [34] and may stem from barriers to access to testing, institutionalized and systemic racism in health care and related medical mistrust in racialized communities, differences

in housing conditions, underlying health conditions or differing occupational exposure risks [35–37]. In Manitoba, 70% of cases to date have been reported to occur in First Nations communities, although these communities make up only 10% of the population [38]. Our findings reinforce the urgent need to investigate contributors to barriers to testing in ethnoculturally diverse communities and to devise community-specific pandemic response planning strategies at the level of government and health policy makers.

Our study has important limitations. First, we were limited to cases confirmed in laboratory testing and could not assess undiagnosed or rapid test-confirmed SARS-CoV-2 infections, though for the period discussed here laboratory testing was employed most often. Income quintile and social determinants of health were assessed at the neighborhood rather than at the individual level, due to privacy limitations in data availability from Statistics Canada. Data on some comorbidities that may be relevant to our outcomes of interest were available only in some provinces and were, therefore, not included in the final analysis. Finally, provinces differed in their testing strategies during the pandemic and experienced "waves" of COVID-19 infection differently, which may have contributed to variability in the aggregate data.

## Conclusions

In this study, we identified a series of demographic, medical, SES, and SDH factors that were correlated with disparities in SARS-CoV-2 testing rates and test positivity. While highlighted by the COVID-19 pandemic, such disparities may reflect underlying barriers to health care access and utilization. Addressing COVID-19 testing inequity in at-risk groups (rural residents, high ethnocultural composition, and lower income) requires unique and dedicated public health initiatives.

## Supporting information

**S1 Table. Database data sources for the provinces of New Brunswick (NB), Manitoba (MB) and Ontario (ON).**
(DOCX)

**S2 Table. Odds ratios and confidence intervals for multivariable regression models of the odds of being tested for SARS-CoV-2, in New Brunswick (NB), Manitoba (MB), and Ontario (ON), including additional covariates for comorbidities and air pollution.**
(DOCX)

**S3 Table. Odds ratios and confidence intervals for multivariable regression models of the odds of testing positive for SARS-CoV-2 among tested individuals, in New Brunswick (NB), Manitoba (MB), and Ontario (ON), including additional covariates for comorbidities and air pollution.**
(DOCX)

**S4 Table. Odds ratios and confidence limits for univariate regression models of SARS-CoV-2 tested individuals in New Brunswick (NB), Manitoba (MB), and Ontario (ON).** Bolded values indicate significance.
(DOCX)

**S5 Table. Odds ratios and confidence limits for univariate regression models of individuals testing positive for SARS-CoV-2 in New Brunswick (NB), Manitoba (MB), and Ontario (ON).** Bolded values indicate significance.
(DOCX)

**S1 Fig. Odds ratio for being tested, univariate models.**
(TIF)

**S2 Fig. Odds ratios for testing positive, univariate models.**
(TIF)

## Author Contributions

**Conceptualization:** Martin Corsten, Stephanie Johnson-Obaseki.

**Data curation:** Lilia Antonova, Chandy Somayaji, Jillian Cameron, Monica Sirski, Maria E. Sundaram.

**Formal analysis:** Lilia Antonova, Chandy Somayaji, Monica Sirski, Maria E. Sundaram.

**Funding acquisition:** Martin Corsten, Stephanie Johnson-Obaseki.

**Investigation:** Chandy Somayaji, Monica Sirski, Maria E. Sundaram.

**Methodology:** James Ted McDonald, Sharmistha Mishra, Jeffrey C. Kwong, Alan Katz, Stefan Baral, Lisa Caulley, Andrew Calzavara, Martin Corsten, Stephanie Johnson-Obaseki.

**Project administration:** Lilia Antonova, Martin Corsten, Stephanie Johnson-Obaseki.

**Supervision:** Martin Corsten, Stephanie Johnson-Obaseki.

**Visualization:** Jillian Cameron.

**Writing – original draft:** Lilia Antonova, Martin Corsten, Stephanie Johnson-Obaseki.

**Writing – review & editing:** Lilia Antonova, James Ted McDonald, Sharmistha Mishra, Jeffrey C. Kwong, Alan Katz, Stefan Baral, Lisa Caulley, Andrew Calzavara, Martin Corsten, Stephanie Johnson-Obaseki.

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
