## [Decision Letter · Decision Letter 0]

12 Oct 2022

PONE-D-22-25812Comparison of socio-economic determinants of COVID-19 testing and positivity in Canada:  a multi-provincial analysis.PLOS ONE

Dear Dr. Antonova,

Thank you for submitting your manuscript to PLOS ONE. After careful consideration, we feel that it has merit but does not fully meet PLOS ONE’s publication criteria as it currently stands. Therefore, we invite you to submit a revised version of the manuscript that addresses the points raised during the review process.

ACADEMIC EDITOR:Please address all of the reviewer's comments.Play particular attention to explaining the study design, and mentioning that three separate analysis was undertaken for each Province.Please describe the statistical approach used, and I would suggest to use a single multivariable regression model instead of several univariable models.

We look forward to receiving your revised manuscript.

Kind regards,

Csaba Varga, DVM MSc PhD

Academic Editor

PLOS ONE

Journal Requirements:

2. Please ensure that you have specified (1) whether consent was informed and (2) what type you obtained (for instance, written or verbal, and if verbal, how it was documented and witnessed). If your study included minors, state whether you obtained consent from parents or guardians. If the need for consent was waived by the ethics committee, please include this information.

"Funding and Disclaimers

This work was supported by Public Health Ontario. This study was also supported by ICES, which is funded by an annual grant from the Ontario Ministry of Health (MOH) and the Ministry of Long-Term Care (MLTC). This study was supported by the Ontario Health Data Platform (OHDP), a Province of Ontario initiative to support Ontario’s ongoing response to COVID-19 and its related impacts. The analyses for Ontario and Manitoba were supported by the Canadian Institutes of Health Research (VR5-172683). This work was supported by the Department of Health and Vitalité Health Network of the Province of New Brunswick under a contract with the New Brunswick Institute for Research, Data and Training at the University of New Brunswick. The results and conclusions are those of the authors and no official endorsement by the Government of New Brunswick or Vitalité Health Network was intended or should be inferred. The study sponsors did not participate in the design and conduct of the study; collection, management, analysis and interpretation of the data; preparation, review or approval of the manuscript; or the decision to submit the manuscript for publication. Parts of this material are based on data and/or information compiled and provided by the Canadian Institute for Health Information (CIHI) and by Ontario Health (OH). However, the analyses, conclusions, opinions and statements expressed herein are solely those of the authors, and do not reflect those of the funding or data sources; no endorsement by ICES, MOH, MLTC, OHDP, its partners, the Province of Ontario, CIHI or OH is intended or should be inferred. J.C.K. is supported by a Clinician-Scientist Award from the University of Toronto Department of Family and Community Medicine. SM is supported by a Tier 2 Canada Research Chair in Mathematical Modeling and Program Science."

6. Please include a copy of Table 1 which you refer to in your text on page 8.

7. Please remove your figures from within your manuscript file, leaving only the individual TIFF/EPS image files, uploaded separately.  These will be automatically included in the reviewers’ PDF.

Reviewers' comments:

Reviewer's Responses to Questions

**Comments to the Author**

1. Is the manuscript technically sound, and do the data support the conclusions?

Reviewer #1: Yes

Reviewer #2: Partly

Reviewer #3: Partly

2. Has the statistical analysis been performed appropriately and rigorously? 

Reviewer #1: Yes

Reviewer #2: I Don't Know

Reviewer #3: No

3. Have the authors made all data underlying the findings in their manuscript fully available?

Reviewer #1: Yes

Reviewer #2: No

Reviewer #3: No

4. Is the manuscript presented in an intelligible fashion and written in standard English?

Reviewer #1: Yes

Reviewer #2: Yes

Reviewer #3: Yes

5. Review Comments to the Author

Reviewer #1: This manuscript examines the differences in COVID-19 testing and likelihood of testing positive for COVID-19 across numerous demographic and social factors in three Canadian provinces. It is extremely well written, clear, concise, yet rigorous in the statistical analysis. I believe it is almost ready for publication consideration after addressing the following:

-I am surprised that the 20-49 age group were most likely to be tested considering the highest risk of severe disease and death among the elderly. Is this associated with limited mobility, access to care, etc? I wonder if you should have included those in long-term care homes (but of course, that could be a separate study...).

-Did you consider adjusting for time since testing effort and availability will have increased substantially since the start of the pandemic (which is the start of your study period)?

-I think it is important to show time series of testing and positivity rates for the 3 provinces.

-Do you suspect these significant relationships may have changed after April of 2021 due to vaccine rollouts, increased access to testing, and the Omicron surges? Why exclude more recent data? Please discuss.

-Can you provide pseudo R squared values for the multivariate models?

-It is not clear if you ran numerous MV regression models or the results in Tables 2 and 3 are one big model, respectively, for each province. Furthermore, was any multicollinearity testing conducted? Might also be good to provide a correlation matrix of your variables.

-You should also discuss the opportunity to conduct spatial regression/analysis since you have a robust dataset that can be aggregated at the neighborhood level. Therefore, you can better identify spatial factors of testing and positivity.

Reviewer #2: Thank you for the opportunity to review this manuscript. Overall this work provides an interesting overview of various factors that may have impacted COVID testing and test positivity in NB, MB and ON. I think that this manuscript could be greatly strengthened by inclusion of additional information, specifically regarding why individual provinces were chosen for inclusion, and additional details regarding the model building process, sufficient for these analyses to be repeated. There appear to be some inconsistencies between model output presented in tables versus within text. Specific comments/suggestions are as follows:

Abstract:

- suggest clarifying within the methods section that data from each province were modelled separately as this is not evident until reading the full methods section.

- it is not clear in the abstract whether variables that were associated with testing or positivity were those from univariate or multivariate analyses - suggest clarifying (e.g., after adjustment for other variables, …)

- the results state that high ethnocultural composition was associated with lower testing, however per Table 2, this is only true across all provinces examined for the highest quintile compared to the lowest - avoid generalizations with respect to trend where results differ across quintiles.

Introduction:

- It is not clear why, when a similar analysis was already completed for Ontario (per Reference 15 - Sunderam et al.), this was largely repeated here rather than any similarities to the published study using Ontario data commented on within the discussion.

- Suggest adding additional context re: why this was repeated (particularly since supplementary tables contain variables included in the original study that are not mentioned or discussed anywhere in this manuscript and that were not included in analyses for other provinces) - presumably this was to allow for a detailed side by side comparison of model output for the different provinces

- It is not clear why MB and NB were chosen for comparison. Suggest adding some additional information regarding the choice of provinces compared - i.e., availability of comparable data, similarities with respect to access to testing, or by random selection of geographically dispersed provinces in Canada.

Methods:

- It is not clear why data were collected for some variables reported in either the methods or supplementary tables (e.g., comorbidities, air pollution) when these are not mentioned in the results/discussion. From the supplemental tables it appears that these were significantly associated with the outcomes of interest and inclusion of these in multivariate models changed the direction of association for some variables (e.g., the odds of individuals aged 85+ being tested relative to those aged 0-4 years).

- There is very little information provided regarding the data modelling approach used. Suggest adding additional information, e.g. if variables were assessed for collinearity, if a backwards approach was used for multivariate models, if model fit was checked.

- Note that Table 1 does not appear to have a title

- Tables are difficult to read - suggest reformatting Tables 2 and 3 similarly to the supplemental tables as CIs are difficult to differentiate from ORs when not presented on separate lines

- Suggest providing some discussion re: why no adjustments were made for potential clustering. It is anticipated that access to testing etc. would further differ between geographic regions within each province.

Results:

- Line 201 states that in all provinces individuals aged 20-34 and 35-49 were most likely to be tested, however per the results presented in table 2, in NB all age groups appeared to have lower odds of being tested compared to 0-4 year olds

- Similarly, in line 206 it states that urban residents were more likely than rural residents to test positive in New Brunswick and Manitoba, however per Table 3 urban residents had higher odds of testing positive in NM and ON, not MB

- Per line 237, areas with higher ethnocultural composition exhibited lower testing rates (Table 2), however per Table 2 this is not universally true, i.e., for both MB and ON, odds of testing were higher in quintile 2 vs. 1 and quintile 3 vs. 1, and in quintile 4 vs. 1 for ON only.

- Suggest commenting on all significant findings, i.e., Table 2 shows that as situational vulnerability increased, there was decreased odds of being tested in MB (with the exception of those in the highest versus lowest vulnerability quintile), while the opposite was true in NB and ON.

- Although not all variables were available for each province, e.g., asthma, air pollution, suggest commenting on these within the text where found to be significant rather than not commenting on these at all.

Discussion:

- Consider providing more information regarding differences between provinces to assist in interpretation of results, i.e., was testing similarly available in each province (i.e., were individuals able to access testing in similar settings, or were there province-specific limitations), also were testing algorithms similar between provinces such that if each individual were to present in any of the 3 provinces would they have been tested? Were test availability and testing indicators consistent across the study period or were there any notable changes that may have influenced results? In ON, individuals working in LTC were required to submit to biweekly testing for a period of time, meaning these individuals may have been more likely to have a positive result identified – this may have influenced rates of testing in certain age groups, e.g. this may partially explain the finding in ON that those aged 20-34 and 35-49 had higher odds of testing relative to those aged 0-4.

- The results of this study are interesting - some additional discussion of findings would be helpful for the reader

- There is no discussion re: why children <5 may have been more likely to be tested than all other age groups in NB

- Why might residential instability be associated with lower positivity?

Reviewer #3: Please see overall comments and details to questions in the track changes in the attached document. The first page contains all information regarding major issues/concerns as well as minor issues. Within the document, the track changes provide more information.

6. PLOS authors have the option to publish the peer review history of their article (what does this mean?). If published, this will include your full peer review and any attached files.

Reviewer #1: No

Reviewer #2: No

Reviewer #3: No

---

## [Author Response · Author response to Decision Letter 0]

11 Jan 2023

Please see attached documents named:

Response to reviewers Dec 03 2022

Revised manuscript with tracked changes Nov 28 2022

Manuscript Nov 28 2022

---

## [Decision Letter · Decision Letter 1]

13 Feb 2023

PONE-D-22-25812R1Comparison of socio-economic determinants of COVID-19 testing and positivity in Canada:  a multi-provincial analysis.PLOS ONE

Dear Dr. Antonova,

Thank you for submitting your manuscript to PLOS ONE. After careful consideration, we feel that it has merit but does not fully meet PLOS ONE’s publication criteria as it currently stands. Therefore, we invite you to submit a revised version of the manuscript that addresses the points raised during the review process.

ACADEMIC EDITOR: One of the reviewer did not feel that the comments were fully addressed.Please address her comments and pay particular attention on addressing the comment related to the description of the modelling process.

We look forward to receiving your revised manuscript.

Kind regards,

Csaba Varga, DVM MSc PhD

Academic Editor

PLOS ONE

Journal Requirements:

Reviewers' comments:

Reviewer's Responses to Questions

**Comments to the Author**

1. If the authors have adequately addressed your comments raised in a previous round of review and you feel that this manuscript is now acceptable for publication, you may indicate that here to bypass the “Comments to the Author” section, enter your conflict of interest statement in the “Confidential to Editor” section, and submit your "Accept" recommendation.

Reviewer #1: All comments have been addressed

Reviewer #2: (No Response)

Reviewer #3: All comments have been addressed

2. Is the manuscript technically sound, and do the data support the conclusions?

Reviewer #1: Yes

Reviewer #2: Partly

Reviewer #3: Yes

3. Has the statistical analysis been performed appropriately and rigorously? 

Reviewer #1: Yes

Reviewer #2: Yes

Reviewer #3: Yes

4. Have the authors made all data underlying the findings in their manuscript fully available?

Reviewer #1: No

Reviewer #2: Yes

Reviewer #3: Yes

5. Is the manuscript presented in an intelligible fashion and written in standard English?

Reviewer #1: Yes

Reviewer #2: Yes

Reviewer #3: Yes

6. Review Comments to the Author

Reviewer #1: I commend the authors for their very thorough and careful responses & revisions. I believe it will be acceptable for publication once the other reviewers are satisfied with their suggested revisions.

Reviewer #2: Thank you for the opportunity to review a revision of this manuscript. I appreciate the authors taking the time to respond to and address reviewer feedback and the manuscript is greatly strengthened as a result. Overall this reads well and provides valuable information for provinces looking to address potential barriers to testing. A few minor final suggestions are provided for consideration for additional reader clarity and study repeatability:

Methods:

• Suggest clarifying how urban versus rural areas were categorized/differentiated and if classification criteria were consistently applied across all 3 provinces

• There is still very little information provided regarding the data modelling approach used. Although collinearity was not identified in the previous study, this was limited to data from Ontario only, and may not be applicable to data from NB or MB.

• Suggest adding a brief overview of the approach used for model building, i.e., if a liberal p-value was used to screen variables from univariate analyses before using a backwards/forwards (as applicable) approach for multivariate models.

• It is still not clear why variables in the final model were not assessed for interaction – one would anticipate that there might be interaction between age and COPD for example. Was the fit of the final models assessed? Were the final variables included in each model the same for each province? (e.g., clarify if any were dropped, or if all were retained, the rationale for doing so)

Discussion:

• Suggest adding some additional context and discussion re: other control measures that were implemented across Canada during the pandemic. While the authors briefly note that restrictions were most stringent in NB, there is no overview or discussion of additional controls (other than the Atlantic bubble), although these undoubtedly contributed to differing opportunities for exposure and subsequent test positivity. With respect to the 3 provinces being compared, these control measures were not consistently implemented across the 3 geographies or for a comparable duration, and were arguably the strictest in NB and the most lax in MB. Less restrictive controls (e.g., less stringent masking mandates, lockdowns, school closures) would have provided more opportunities for community exposure, and could have contributed to the differing test positivity observed here.

• With respect to equity in access to testing, do you think introduction and distribution of rapid tests would have helped to address the observed inequity?

Reviewer #3: Please find my comments and edits in the attached document. Note that there are track changes as well. Thank you!

7. PLOS authors have the option to publish the peer review history of their article (what does this mean?). If published, this will include your full peer review and any attached files.

Reviewer #1: No

Reviewer #2: No

Reviewer #3: No

---

## [Editor Report · Decision Letter 2]

17 Jul 2023

Comparison of socio-economic determinants of COVID-19 testing and positivity in Canada:  a multi-provincial analysis.

PONE-D-22-25812R2

Dear Dr. Antonova,

We’re pleased to inform you that your manuscript has been judged scientifically suitable for publication and will be formally accepted for publication once it meets all outstanding technical requirements.

Kind regards,

Csaba Varga, DVM MSc PhD

Academic Editor

PLOS ONE
---

## [Editor Report · Acceptance letter]

14 Aug 2023

PONE-D-22-25812R2 

Comparison of socio-economic determinants of COVID-19 testing and positivity in Canada: a multi-provincial analysis. 

Dear Dr. Antonova:

I'm pleased to inform you that your manuscript has been deemed suitable for publication in PLOS ONE. Congratulations! Your manuscript is now with our production department. 

Kind regards, 

on behalf of

Dr. Csaba Varga 

Academic Editor

PLOS ONE